# T-Cell Receptor Repertoire Sequencing and Its Applications: Focus on Infectious Diseases and Cancer

**DOI:** 10.3390/ijms23158590

**Published:** 2022-08-02

**Authors:** Lucia Mazzotti, Anna Gaimari, Sara Bravaccini, Roberta Maltoni, Claudio Cerchione, Manel Juan, Europa Azucena-Gonzalez Navarro, Anna Pasetto, Daniela Nascimento Silva, Valentina Ancarani, Vittorio Sambri, Luana Calabrò, Giovanni Martinelli, Massimiliano Mazza

**Affiliations:** 1IRCCS Istituto Romagnolo per lo Studio dei Tumori (IRST) “Dino Amadori”, 47014 Meldola, Italy; lucia.mazzotti@irst.emr.it (L.M.); anna.gaimari@irst.emr.it (A.G.); sara.bravaccini@irst.emr.it (S.B.); roberta.maltoni@irst.emr.it (R.M.); claudio.cerchione@irst.emr.it (C.C.); valentina.ancarani@irst.emr.it (V.A.); luana.calabro@unife.it (L.C.); giovanni.martinelli@irst.emr.it (G.M.); 2Platform of Immunotherapy HSJD-HCB, Immunology Service, August Pi Biomedical Research Institute, Sunyer, Hospital Clinic de Barcelona, 08036 Barcelona, Spain; mjuan@clinic.cat (M.J.); eagonzal@clinic.cat (E.A.-G.N.); 3Department of Laboratory Medicine, Karolinska Institutet, 17177 Stockholm, Sweden; anna.pasetto@ki.se (A.P.); daniela.silva@ki.se (D.N.S.); 4Microbiology Unit, The Great Romagna Area Hub Laboratory, 47522 Pievesestina, Italy; vittorio.sambri@unibo.it; 5DIMES, Bologna University, 40126 Bologna, Italy

**Keywords:** TCR repertoire, TCR sequencing, infectious diseases, cancer immunotherapy, HLA, TILs, COVID-19, T-cell response

## Abstract

The immune system is a dynamic feature of each individual and a footprint of our unique internal and external exposures. Indeed, the type and level of exposure to physical and biological agents shape the development and behavior of this complex and diffuse system. Many pathological conditions depend on how our immune system responds or does not respond to a pathogen or a disease or on how the regulation of immunity is altered by the disease itself. T-cells are important players in adaptive immunity and, together with B-cells, define specificity and monitor the internal and external signals that our organism perceives through its specific receptors, TCRs and BCRs, respectively. Today, high-throughput sequencing (HTS) applied to the TCR repertoire has opened a window of opportunity to disclose T-cell repertoire development and behavior down to the clonal level. Although TCR repertoire sequencing is easily accessible today, it is important to deeply understand the available technologies for choosing the best fit for the specific experimental needs and questions. Here, we provide an updated overview of TCR repertoire sequencing strategies, providers and applications to infectious diseases and cancer to guide researchers’ choice through the multitude of available options. The possibility of extending the TCR repertoire to HLA characterization will be of pivotal importance in the near future to understand how specific HLA genes shape T-cell responses in different pathological contexts and will add a level of comprehension that was unthinkable just a few years ago.

## 1. Introduction

The immune system guards multicellular living organisms from the damage of pathogens [1].

Adaptive immunity defense is elicited by large numbers of T-cell receptors (TCRs) and B-cell receptors (BCRs) [1], and perception and adaptation to external insults enhance pre-existing and generate de novo receptors that the immune system records and retains as immunological memory (e.g., with vaccination) [2].

T-cells, the main actors in cytotoxic immune responses and also contributing to the full activation of the humoral adaptive response, include mainly αβ T-lymphocytes expressing TCR alpha and beta chains and γδ T-cells, a smaller population expressing TCR gamma and delta chains; in humans, these different types of T-cells are about 95% and 1–5%, respectively [1,3,4].

The αβ T-cells are essential in cellular immunity since they mediate recognition of antigen peptides in a major histocompatibility complex class I and II (MHC) restricted manner, defining antigen complexes and driving the antigen-specific adaptive immune response against non-self-perceived antigens, including pathogens and cancer neo-antigens, through the TCRs [5].

The TCR is a heterodimeric plasma membrane protein located on T-cells, and it is composed of two paired chains [6,7] whose loci are organized as gene segment families comprising a variable (V) gene, a diversity (D) gene, a joining (J) gene and a constant (C) gene.

During T-cell maturation in the thymus, the β and δ chains’ VDJ gene fragments are rearranged as one allele of each gene segment randomly recombines with the others, composing a functional V segment [8], while, for α and γ chains only, VJ fragments recombine as there is no D gene [4]. Each coding segment is flanked by conserved DNA regions, the recombination signal sequences (RSSs), which serve as cleavage signals for enzymes encoded by recombination activating genes RAG1 and RAG2. After RSSs recognition, these enzymes introduce double strand breaks (DSBs), initiating the V(D)J recombination process [9,10].

The V region of the α and β chains has three hypervariable complementary-determining regions (CDRs): CDR1 and CDR2 come with the V gene and play a role in both the interaction and stabilization of the TCR-MHC complex [8], while the most variable CDR3 is encoded by VDJ or VJ segments [3] and determines most of the binding specificity of the TCRs to the antigen-MHC complex [8,11].

In addition to somatic rearrangements (recombination diversity), insertions or deletions may occur randomly at the junctions of both VD and DJ fragments (junctional diversity) for a functional TCR sequence [3,12]. As the recombination mechanism just described occurs separately for both chains, the combinatorial diversity increases even further the number of possible TCR variants [13] (Table 1).

T-cells activation is triggered by an antigen-specific signal and a co-stimulatory cue due to the CD28 binding by T-cells to B7 ligands on the antigen presenting cells (APCs) [15]. Following their activation in the periphery, T-cells start to proliferate and undergo clonal expansion [4], generating a population of expanded T-cell clones (clonotypes) that share the same TCR sequences conserved during mature T-lymphocytes mitosis [15].

TCRβ clonotypes in individuals are estimated to 10^6^–10^8^ [16,17] on a total estimated number of 10^12^ circulating lymphocytes [18]. Therefore, even if the actual diversity of the paired TCRα and TRBβ repertoire is still debated and estimated to range from a lower limit of 10^15^ up to 10^61^ [14], is it possible to assay an overall complexity of 10^4^–10^6^ T-cells only in an individual experiment [19]? At present, no technologies are available and sufficiently empowered to describe such an immense source of variability.

The entire amazingly broad and diverse assembly of TCR sequences, known as the TCR repertoire, is composed of naive T-cells, antigen-inexperienced cells shaped in the thymus, where new antigen specificities are introduced [20,21], and memory T-lymphocytes, antigen-experienced cells whose specificity depends on antigen exposure through people’s lives and persisting long-term [22].

The TCR-MHC affinity can delineate the T-cells propensity, defining the level of equilibrium between effector and memory T-cells [23,24].

Naive and memory T-cells make up a unique TCR footprint that is different even in genetically identical twins [25] due to the intrinsically stochastic nature of TCR generation and immune system personal experience.

Moreover, the TCR repertoire footprint dynamically evolves according to the challenges with which the immune system is confronted [4], such as infections, aging, autoimmune diseases, cancer and many other stimuli. Importantly, the exposure to an antigen triggers a massive expansion of antigen-specific T-cells, altering the composition of the TCR repertoire in favor of specific clonotypes (TCR bias) [26,27] that can be correlated to a biological process and exploited as “molecular barcodes” both in health and diseases [15].

## 2. TCR Repertoire Analysis

It allows the definition of these types of “barcodes” in different contexts, revealing, for example, important information about a successful antitumor T-cell response, on how to improve efficacy and safety of immune checkpoint inhibitors (ICI), on the tumor microenvironment (TME) characterization, on the immune response during disease development and treatment, on minimal residual disease (MRD) assessment, transplantation, autoimmune disease and infectious disease characterization and therapy [1,4,8,13,28,29,30].

In this review, we describe TCR repertoire sequencing strategies and applications (Figure 1) in individuals exposed to infectious agents, such as HIV, HBV, HCV and SARS-CoV-2, and to cancer, with an updated overview of the available technologies.

## 3. TCR and HLA

As already mentioned, T-cell activation occurs as a consequence of the specific recognition between TCR and foreign antigen peptides presented by the MHC molecules (Figure 2), which are transmembrane glycoprotein complexes expressed on the cell surface. 

MHCs in humans are coded by the highly polymorphic human leukocyte antigen (HLA) gene family located on chromosome 6 and involved in the identification of self versus non-self.

Three subclasses of HLA molecules are expressed in various human tissues: HLA class I, class II and non-classical HLA molecules; some of them make up the class III region.

All nucleated cells express HLA class I proteins on the plasma membrane, allowing to expose peptides derived from intracellular antigens to CD8 T-cells monitoring via TCR interaction. As a result, cells expressing viral or mutated non-self-antigens are killed directly to restrain infection and prevent further cell transformation [31]. There are three main HLA types within this class encoded by the HLA-A, HLA-B and HLA-C loci. 

HLA class II proteins are constitutively expressed by professional APCs, including B-cells, and their expression can be upregulated on activated immune cells, binding peptides derived from antigens captured from outside of the cell, presenting ‘exogenous’ peptides to CD4 T-cells [31]. The three main types of HLA in class II are encoded by the HLA-DR, HLA-DQ and HLA-DP loci. APCs can present outside captured antigens also using MHC class I through a process of cross-priming/cross-presentation.

More than 30,000 HLA variants among class I and II have been determined so far, and their permutations raise the number of possible combinations to astronomical numbers, making it unlikely that the individual’s resulting HLA type would be shared with an unrelated individual and defining the subset of peptide epitopes that could be presented for immune surveillance [32].

The class III alleles encode for factors involved in the inflammation process, leukocytes differentiation and the complement system [33]. 

In addition to HLA proteins’ role in the human immune system T-cell activation, HLA type plays an important role in driving T-cell positive and negative selection in the thymus, thereby also shaping the naive T-cell repertoire.

HLA importance is evident in the context of organ and bone marrow transplantation, being responsible for the rejection process, but many studies today link the HLA type with disease susceptibility or development and response to therapy for many diseases.

The first link found in this context has been the discovery of HLA-B and Hodgkin lymphoma association [34], and, since then, MHC is considered the genome region with the greatest amount of association with human diseases [35] (some examples are shown in Table 2).

An example of complemented data can be found in De Witt III et al.; in their work, they analyzed TCRs from a cohort of 666 healthy volunteer donors to find links between TCRs profiling and HLA associations to disease [32]. Starting from the analyses of the common TCRs across the whole cohort, the study of TCR-HLA association patterns co-occurrence, as a strong influence by HLA alleles distribution, was observed in accordance with the fact that most αβ TCRs are HLA-restricted. 

Additional analyses revealed that significant TCR clusters, shared within the cohort, may represent markers of immunological memory and showed that most highly HLA-associated TCRs are related to common viral infections, such as influenza virus and Epstein–Barr virus (EBV).

Moreover, they further analyzed CDR3 sequence–HLA allele correlations, identifying a significant negative association between CDR3 and peptide charges, which suggests that the maintenance of charge complementarity across the TCR-MHC complex is a relevant feature of binding.

These results demonstrate the potential of combining statistical tools to TCR repertoires and immune exposure as sequences from the clusters can infer a TCR expansion driver.

Thus, TCR sequence–disease associations are complicated by individual HLA type dependence, thereby characterizing the TCR-HLA interactions. Therefore, it is crucial to understand antigen discrimination by T-cells and to deepen our comprehension of the interplay and associations among individual HLA type, TCR sequences and disease. In this respect, the implications for the development of novel therapeutics are obvious and find translation to many disease settings, including infectious diseases, autoimmune diseases and oncology.

## 4. TCR Repertoire via HTS: When Details Matter

High-throughput sequencing (HTS) has emerged as a suitable method for evaluating TCR diversity, allowing the characterization of immune repertoires with massive parallel sequencing at a deeper and finer level [8,42]. This technique combines the resolution of individual TCR nucleotide sequences decoded with the ability to read millions of sequences simultaneously [43]. Traditional strategies, such as spectratyping, Sanger sequencing and other assays, such as flow cytometry [30], are time-consuming and insufficient for generating a deep analysis of the immune repertoire.

To perform a TCR repertoire analysis, many aspects must be taken into account, such as the kind of starting material for the library preparation, the method for sequencing [8] and the following data analysis pipeline. 

First, following a nucleic acid extraction from the samples cohort of interest, genomic DNA (gDNA) and messenger RNA (mRNA) can both be used for library preparation [44].

The amount of gDNA is proportional to the number of analyzed cells with a 1:1 number of clonotypes and number of cells ratio (1 gDNA template per cell), allowing us to determine the relative abundance of sequences in a sample at the cost of unavoidably detecting potentially irrelevant and non-expressed sequences that must be removed through post-processing bioinformatic analysis [45,46].

On the contrary, mRNA is related to cell function/activation [1], and RNA-based methods are more sensitive due to the presence of multiple copies of the transcript of interest per cell. Thus, a more comprehensive recognition of both unique receptor variants and functional expressed TCRs can be obtained using RNA as it allows the detection of very rare clones and reveals sequences effectively transcribed and thus more likely to yield functional TCRs [7,45].

Further, gDNA as input material does not require the reverse transcription step, minimizing the possible biases introduced in cDNA synthesis [5], while starting from already spliced mRNA converted into cDNA holds the advantage that less reverse primers are sufficient for C region amplification, reducing PCR biases from multiplexed J primers [30,42], obtaining both a higher detection sensitivity and no need for adapter sequences [47].

Additionally, RNA-based methods allow the implementation of unique molecular identifiers (UMIs), which consist of random DNA sequences added during cDNA synthesis in order to label individual cDNA molecules, correcting for amplification and sequencing errors [48]. However, since RNA-based approaches are affected by the relative expression of TCRs in the cells and not only by the number of cells expressing the same TCR, those methods are believed to be less reliable in describing the relative abundance of clonotypes in a cell population. The advantages and disadvantages of strategies based on gDNA and mRNA are listed in Table 3.

The choice between gDNA and mRNA as a source for TCR repertoire sequencing depends on the quantity of nucleic acids requested to start a specific workflow, on the options for library preparation and on the type of results required at the end of the process. Currently, peripheral blood is the most used starting material due to the ease and non-invasive sampling procedure, especially in relation to cohorts of healthy subjects, even if peripheral blood lymphocytes are estimated to 2% only of the total lymphocytes in the body [1].

To obtain robust and comparable data, it is important to standardize the processing of all samples to process them as uniformly as possible, starting from a determinate amount of material concentration for each sample and analyzing them with the same parameters as a comparable number of reads in relation to the depth of analysis to achieve in the experiment [53]. 

The sequencing depth can be adjusted according to the sample type and experimental goals. Deeper sequencing is appropriate when analyzing samples with large or diverse cell populations at the expense of higher throughput.

The majority of TCR repertoire profiling studies are based on the analysis of the CDR3 region; however, full-length sequencing includes additional regions, such as CDR1 and CDR2, involved in antigen receptor binding affinity and/or downstream signaling, and allows to directly clone and express the identified and chosen receptors to perform others experiments. This aspect is crucial when the identification of therapeutic candidate TCRs is the goal of the analysis [45].

TCR HTS methods can be divided into bulk sequencing, for T-cell populations evaluation or single-cell sequencing for the analysis of individual T-cells [2]. The choice between these two analyses depends on the goal of the experiment and on other factors, such as sample requirements, hands-on and total workflow time, degree of polymerase chain reaction (PCR) bias, quantifiability, immune repertoire coverage, ease of data analysis and cost. Generally, for the analysis of immune repertoire diversity in health and disease, a bulk sequencing approach is used because it allows the sampling of many more sequences in a single experiment, even if information about αβ-TCR pairing is lost and undetected low-frequency TCRs could mislead diagnostics outcomes [54]. 

However, a single-cell approach is preferred in experiments set to investigate the specificity of a TCR for an antigen of interest, for capturing the paired αβ-chains information and producing complete antigen receptors and/or characterizing their function. 

Using mRNA, single-cell TCR sequencing makes it possible to evaluate cell transcriptional heterogeneity down to the single nucleotide level and gene expression variability at the single cell level, leveraging the study of phenotypically different cells populations to an unprecedented resolution [55,56]. Compared with TCR bulk sequencing, the number of cells sequenced using the single-cell approach drops to 10^2^–10^3^ instead of up to 10^6^ [54], and another consideration to take into account is that isolating single cells can be challenging, and obtaining viable cells at the end of the process requires care and to work quickly, with a consequent decrease in the number of samples analyzed and an increase in variability in any single-cell study due to the process workflow. If starting from cryopreserved cells, it must be tested whether the process of cryopreservation has changed or damaged the cell viability and/or phenotype. Moreover, data analysis requires specific tools and expertise regarding the most appropriate analytical approaches. Because of the above-mentioned considerations, single-cell is still a more expensive method compared to other sequencing techniques.

Researchers have often performed initial bulk analysis and moved, after selection of features of interest, such as binding affinity, to single-cell [45], even if recently developed commercial single-cell sequencing solutions start to provide full-length paired αβ-chains sequencing of many T-lymphocytes [2]. These ultimate technologies are based, for example, on barcoded gel beads mixed with cells, enzymes and partitioning oil, used to generate V(D)J gene expression libraries (e.g., 10X Genomics, Pleasanton, CA, USA). A major improvement in the throughput came with emulsion-based approaches [7] in which single cells are encapsulated in water-in-oil emulsions, where cDNA is synthetized thanks to TCR primers and RT-PCR reagents and then sequenced [5], maintaining native αβ-chains pairing while sequencing both chains. Quantitative transcriptomics is used to analyze TCRs and other cell markers, and since each cell is individually barcoded, amplification bias is not an issue. Using the same principle of the methods reported above, microfluidic platforms reliant on individual cell compartmentation in microwells or droplets have been applied in single-cell isolation [57].

Droplet-based instruments require a dedicated hardware platform [58], and encapsulation efficiency is variable depending on which method is used, as thousands of cells can be encapsulated, but rare clones can still be missed, while costs remain elevated and hinder a broad application of those approaches [59]. To improve single-cell TCR sequencing, the cells sorting through the FACS instrument represents an approach to enrich populations of interest through surface markers presence and helps in analyzing rare subsets of T-cells [5].

Library construction and data analysis of bulk and single-cell sequencing approaches share essentially the same principles and workflows [60].

Multiplex PCR represents the most used approach to prepare sequencing libraries for TCR repertoire analysis, as emerged in Appendix A, and consists of two rounds of PCR using multiple primers, specifically a set of forward primers for V genes and a set of reverse primers for either J or C genes, according to the template used [5].

At first, receptor locus amplification takes place with the addition of known sequences; all the possible recombination events of the receptor sequences are captured using a V and J primers pool in the first PCR, while the additional sequences are fundamental for the incorporation of sequencing adaptors and indexes to each amplicon during the second PCR. By the way, multiple rounds of PCR before sequencing could introduce sequencing biases due to the fact that the priming sequences at the 3′ and 5′ ends of the first PCR overlap significantly between different sites, implying the use of a pool of slightly diverse primers for different TCR sequences amplification, and, importantly, receptor sequences that share similarity with the primers used could be recognized more effectively, affecting clone frequencies’ results, with a negative impact on research outcomes [45]. However, it is possible to quantify templates before and after multiplex PCR using synthetic TCR molecules targeted by the multiplexed primers pool, with primer concentration optimization and correction of potential biases [61], and many assays commercially available for library preparation already contain validated internal controls that correct for these biases. 

Rapid amplification of cDNA ends is an approach of library preparation used only for RNA templates and relies on a template switching mechanism, an intrinsic property of certain reverse transcriptases (RTs) [62]. This method avoids amplification bias between V regions introduced by multiplex PCR [47], and it is applicable both to 5′ or 3′ ends. The 3′ RACE approach takes advantage of the mRNA poly(A) tail at the 3′ end by using it as a generic priming site for the PCR amplification step following retrotranscription, targeting the region of interest between a known exon and the 3′ end [63].

The 5′ end of mRNA does not present any generic priming sites; therefore, accurate incorporation of an adapter sequence at the cDNA first-strand 5′ end, by adding non-templated nucleotides through RT activity, is required. A hybridization step occurs then between a template-switching oligo complementary to the added non-templated nucleotides, enabling templates to switch and enabling adapter sequence incorporation, which serves the next two PCRs [45]. Consequently, targeting the 5′ adaptor sequence and the C region by using just one pair of primers is enough for all TCR rearrangements’ amplification [5], thus reducing PCR errors and ensuring the TCR repertoire profile matches the original sample instead of the primer design. Additionally, a high-on-target rate is guaranteed by the second semi-nested PCR, with a decrease in sequencing costs [45].

Some library prep protocols still employ ligation reaction to anchor adapters and barcodes to the amplicons even if the suboptimal ligation efficiency of the adapters could represent a limiting factor of this choice [62,63], impacting the accuracy of the quantification, especially for the low frequency TCRs, which can justify why 5′ RACE is less reproducible than multiplex PCR [8,64]. 

After finishing the library preparation step, sequencing of samples can be continued, most of which are run on Illumina platforms. TCR sequences obtained at the end of the workflows consist of sequences of nucleotides that have to be first aligned to VDJ regions’ reference sequences and then grouped according to sharing the same CDR3 in order to evaluate clonotypes [65].

The use of algorithms, such as IgBLAST [66], IMGT/HighV-QUEST [67], MiXCR [68], immuneSIM [69] and RTCR [70], allow the evaluation of TCR sequences analogies and discrepancies as compared to publicly available TCR databases.

TCR repertoire analysis is, nowadays, becoming more and more accessible to the scientific community and the pharma industry to unravel TCR specificities, clonality, diversity and the intensity of response associated with treatments and disease states. The panorama of applications for TCR sequencing on the market is really broad and complex, with many companies proposing specific protocols according to the specificities previously defined. As a means of orienting researchers in the choice of the best suited approach for TCR sequencing in the present scenario we provide, to the best of our knowledge, a comprehensive and synthetic description of the kits and the services accessible on the market at the moment is provided (Appendix A).

## 5. TCR and Infectious Diseases

During a viral infection, CD8+ T-lymphocytes kill virus-infected cells with the help of T-helper 1 (Th1) cells, a population that derives from CD4+ T-cells, together with another population of T-follicular helper (Tfh) cells, which promotes somatically hypermutated antibody generation from B-cells. Other functions mediated by CD4+ T-cells are the enhancement of NK cells and macrophages’ action through cytokine release [71,72,73], the production of chemokines from infected tissues and the recruitment of effector cells [74]. In addition, Th1 cells exert direct cytotoxic functions against infected cells in an MHC II-restricted manner [75,76].

Human immunodeficiency virus (HIV) infection, which is characterized by a first stage (~6 weeks) of rapid viral replication in infected cells followed by an 8–10 years asymptomatic period [77], represents an example of the importance of CD4+ T-cells’ role.

During this prolonged phase, latent pro-virus is established in multiple tissue compartments and maintained by a low-level of replication. This consequent chronic immune activation drives progressive and further depletion of CD4+ T-cells [78,79] and is associated with an increasing dysfunction in CD8+ lymphocytes.

These processes result in T-cell exhaustion with upregulation of several inhibitory receptors [80,81], causing increased susceptibility to a broad spectrum of pathogens [82].

The reduction in CD4+ T-cell counts, coupled with an increase in CD8+ T-cells, leads to an inversion of the CD4+:CD8+ ratio and typically predicts the non-acquired immunodeficiency syndrome (AIDS) morbidity [83].

This CD4+:CD8+ ratio inversion could be partially rescued by the administration of antiretroviral therapies (ART) that inhibit viral replication, CD4+ T-cells progressive loss and AIDS development in most infected individuals, although they do not lead to complete viral clearance [84].

Most HIV-infected cells are characterized by central and effector memory CD4+ T-cell phenotypes, and their clonal expansion is maintained by latently infected cells that can expand and contract over time, similar to the behavior shown by adaptive immune responses, keeping the reservoir relatively stable for long periods [85,86,87,88,89,90,91,92,93].

AIDS progression is associated with the TCR repertoire diversity depletion [94,95].

Deep sequencing has enabled the characterization of immune responses against HIV [96,97] by decoding the antiviral clonotypes and by exploiting those found to be protective for vaccination and for the development of novel immunotherapies [98].

Shared clones from selected cohorts of patients can also be used as multiparametric biomarkers for diagnosis and disease monitoring [99,100].

In this regard, it has been demonstrated that an HIV DNA vaccine was able to elicit a similar response in terms of shared clonotypes to that of HIV controllers, patients who naturally control the infection [98], suggesting that the vaccine administration can induce high-affinity CD4+ T-cell responses [101].

Virus-specific clonotypes can be used to track the amplification of latently infected CD4+ T-cell clones, demonstrating that homeostatic CD4+ T-cells proliferation is the main factor responsible for HIV provirus persistence and evaluating the TCR repertoire profile in patients after treatment and therapeutic approaches to follow the evolution of the anti-viral response [102].

In their study, Wanjalla et al. combined both bulk and single-cell RNA sequencing to identify CD4+ T-cells clonality in AIDS patients’ adipose tissue with or without diabetes and an HIV-negative diabetics’ control group, hypothesizing a role of latent pro-viral and/or replicating HIV and other HIV-associated viruses, such as CMV, in the development of glucose intolerance [103]. They demonstrated that the recruitment and proliferation of pro-inflammatory and cytolytic virus-specific CD4^+^ T-cells may alternate adipocyte function and increase the metabolic disease risk in HIV patients, providing an example of using TCR characterization to correlate different clinical conditions [103].

Pilkinton et al. applied sequencing in a cohort of chronically HIV-infected ART-naive individuals for investigating the Gag-specific CD4+ and CD8+ TCR repertoires determining the relationship between HIV-specific TCR repertoire diversity and HIV sequence variation in that cohort of patients. Firstly, they stimulated T-cells with HIV Gag peptide and sorted them based on the presence of surface activation markers CD69 and CD25 [104,105,106,107]. The sorted cells were then sequenced to assess T-cell diversity in the CD4+ and CD8+ compartments, demonstrating a significantly higher TCR diversity of Gag-reactive CD4+ T-cells than that of CD8+ T-cells, with few highly expanded clonotypes. These results have been confirmed by other recent studies on non-antigen-specific CD4+ and CD8+ T-cells in which healthy adults have five times higher CDR3 diversity in the CD4+ compartment, and this relationship is consistent regardless of individuals’ age [108,109].

From these studies, an inverse association between TCR repertoire diversity and viral diversity can also be underlined, suggesting that a major clonotype differentiation may better limit virus adaptation, resulting in HIV infection control, as would be expected, as having an immune system able to deal with multiple epitopes presented by HIV-infected T-cells correlates with decreased viral load [110,111,112,113,114], although the order of magnitude of TCR diversity required to exert optimal control of HIV infection has not been defined so far [107].

Similarly, those considerations and studies can be applied to other viruses, such as those belonging to the Hepadnaviridae family, such as Hepatitis B virus (HBV) and Hepatitis C virus (HCV).

HBV causes acute infections in humans, which can last from 8 weeks to over 6 months, becoming a chronic infection in those individuals whose immune system fails to tackle the virus due to the exhaustion of T-cells and other immune cell dysfunctions [115].

Deep sequencing has been used to characterize the immune repertoire in HBV infection, underlining the presence of shared and immunodominant clonotypes associated with the development and treatment of the disease and the role of CD8+ T-cells in the pathogenesis of the chronic phenotype with the expression of specific and predominant TRB V families usage; an updated list of these associations has been recently reported [115].

Recent studies also aim to apply the TCR sequencing in order to discriminate the population that responds to the HBV vaccine versus the 5–10% of people that are unable to respond and to investigate specific CDR3 motifs to be used as potential novel targets in the designing of HBV vaccines [116].

Likewise, recent studies identify CD8+ TCR clonotypes unique to HCV that expanded during acute infection and reinfection [117] and demonstrate the feasibility of engineered T-cells development against HCV based on the use and expression of isolated and well-characterized anti-HCV specific TCRs [118].

Adaptive immune responses are crucial to restrain and clear severe acute respiratory syndrome coronavirus 2 (SARS-CoV-2) infection and determine patients’ clinical outcomes as early immune responses primarily play a protective role [119]; in fact, adequate T-cell counts and clonal expansion are evident in COVID-19 convalescent patients [120,121]. However, dysregulated and exaggerated inflammatory responses can fail in clearing the virus and lead to severe symptoms and to fatal disease outcomes [122].

COVID-19 patients’ data mortality indicate age, male gender and pre-existing comorbidities as risk factors associated with an increased fatality rate [123,124], and the total lymphocytes count drop observed in non-surviving COVID-19 patients suggests that lymphocytes quantitation in circulation might be a predictive biomarker for the severity of this disease [125,126,127].

Both CD4+ and CD8+ T-lymphocytes are characterized by high expression of genes involved in the inflammatory process in severe COVID-19 patients [127].

Some studies showed changes in immune cell composition [128,129], cell–cell communication [130,131] and gene expression [132,133] in COVID-19 patients [134]; additionally, the application of deep sequencing for decoding the TCR repertoire of these patients may explain the clinical outcome disparity observed [22].

The immune repertoire of COVID-19 patients infected by SARS-CoV-2 was reported in 2020 [135] and characterized for decreasing in the early stage of the disease while increasing during the recovery period, with T-cell response peaks about one to two weeks after infection lasting during the months of recovery [136]. 

In the context of the COVID-19 pandemic, the TCR repertoire has been and still is extensively studied through the HTS approach, with the aim to investigate the T-cell clonal composition at different stages of infection, for example, by comparing patients in the acute phase with patients in the convalescent phase or with healthy controls, to identify clonotypes that are specific after exposure to the virus and maybe identify the protective ones or correlating T-cell responses with serological levels of anti-SARS-CoV-2/neutralizing antibodies and with the clinical outcome of patients. This approach also allows to identify clones originated de novo after infection or after vaccination with different schemes or types of vaccine schemes. Single-cell sequencing has been used to compare the TCR repertoire between COVID-19 patients and healthy controls, observing a very significant reduction in TCR clones diversity and disease-related changes in V and J gene usage, with less gene subfamilies and preferential V-J combinations used in COVID-19 patients. Those findings have been confirmed by others [137,138], specifying that the preferential V(D)J sequences only exist in the convalescent patients [137]. 

With reference to this latter group, the authors investigated TCRα and TCRβ composition in CD4+ and CD8+ T-cells, studying the characteristics of the TCR repertoire across all convalescent stages. Their results show that TCR clonality, for example, is higher in CD8+ as compared to CD4+ T-cells.

Those observations are also confirmed by other studies performed in patients with severe infection where CD8+ T-cells were less expanded, more proliferative and more phenotypically heterogeneous [128], while CD4+ T-cells had a higher rate of cytotoxic granules release and a marked decrease in the secretion of functional molecules [127].

Some shared TCRs sequences appeared 2 weeks after convalescence and persisted up to 6 months, suggesting these clones might represent the T-memory phenotype associated with the disease [137]. The same observation has been reported in a population of antibody-seronegative convalescent individuals with asymptomatic and mild COVID-19, where the presence of SARS-CoV-2-specific polyfunctional stem-like memory phenotype T-cells in the peripheral blood suggests that an effective T-cell response may exert sufficient immune protection against the virus, even in the absence of neutralizing antibodies [127,139]. This aspect might be particularly relevant for patients affected by B-cell deficiency and highlights the inadequacy of considering only serological measurements as a surrogate of protection in vaccinated or exposed individuals. 

Another suggestion from Y. Wang and colleagues is that cross-reactive CD4+ and CD8+ T-cells lymphocytes contribute to SARS-CoV-2 infection resistance via the production of a virus-specific TCR recombination pattern [137] as overlap indices of TCRB CDR3 amino acid sequences are higher in COVID-19 patients. Those data are now helping to fingerprint a shared clonal expression responsible for the anti-SARS-CoV-2 immunity within a population of COVID-19 patients and of vaccinated individuals, which are protected by the most adverse events related to COVID-19. 

An extensive effort was undertaken by several institutions around the world, including us, to establish the ImmuneCODE™, a comprehensive database of the T-cell response to SARS-CoV-2 infection [136,140]. To compile the ImmuneCODE, the TCRB repertoire from 1014 subjects exposed to, suffering from or recovered from COVID-19 was deeply analyzed. In addition, MIRA technology, which can pair TCRs to their cognate antigens, identified 150,000 high-confidence SARS-CoV-2-associated TCRs from exposed subjects compared with naïve controls [140].

The ImmuneCODE data™ have been made publicly available and can be downloaded and analyzed to better understand the immune response to the SARS-CoV-2 virus and develop new strategies and clinical trials against COVID-19. Based on ImmuneCODE, the immunoSEQ^®^ T-MAP™ COVID was developed, which consists of a T-cell-based clinical test to quantify SARS-CoV-2-specific T-cell immune responses, and, to the best of our knowledge, this is the first example of a commercially available tool where the TCR repertoire can be employed to monitor the response to a pathogen in a TCR sequence-specific manner down to the clonal level.

This approach paves the way for the use of the TCR repertoire as a novel complex biomarker to track the exposure of individuals to non-self-antigens.

## 6. TCR and Cancer

Cancer is derived from a series of genetic alterations that occur in normal cells that are transformed into malignant cells [141], capable of invading the surrounding normal tissue and generating metastasis.

The onset of malignant clones is constantly monitored and countered by lymphocytes, in particular by cytotoxic T-cells recognizing tumor specific/associated antigens through their TCRs.

Mutations in affected cells can lead to the expression of immunogenic mutant proteins, the so-called neoantigens, that can be targeted by the immune system. Since neoantigens are not expressed by normal cells, they can be found utilizing genomic deep sequencing of tumor tissues and can represent attractive targets for cancer therapy [141,142]. Importantly, overcoming the peripheral immune tolerance can lead to targeting also of self-antigens, which can be overexpressed in specific cancer subtypes [143]. If diagnosed at early stages, most cancers can be cured, but the levels of cancer-related biomarkers are low for the early onset of the disease, making it very tricky the detection of cancerous cells using traditional methods, leading to a delay in the diagnosis, to higher stage and aggressivity of neoplasia and lower chances of cure.

Strategies using immune repertoire deep sequencing coupled with bioinformatic analyses were beneficial in identifying antigen-specific sequences [1], even if predictive biomarkers to establish the patient’s immune status are still missing [65]. 

TCR repertoire analysis can: (i) describe important aspects of the tumor microenvironment (TME), (ii) intercept changes in immune responses during the disease and of treatment, (iii) be markers of responders to immunotherapy and (iv) be used to identify candidate immunogenic neoantigens for vaccine design [3,8,13,29,30].

The TME is the environment where the tumor exists and corresponds to a complex system of various intercommunicating cell types, including cancer cells, immune cells, stromal cells and others [3,144], whose influence makes T-cell repertoire distribution in tumors, normal tissues and peripheral blood heterogeneous [145].

The use of deep sequencing allows to profile the heterogeneity of the TME [146], enabling a more precise investigation of cancer evolution, of local interactions among cells and of immune regulations specifically happening in the tumor milieu, strengthening our understanding of cancer pathogenesis and immune suppression [147,148]. It is now accepted that the TME is targetable for tumor therapy [149,150] since stromal cells are genetically stable and promote tumor growth through phenotypic changes [151].

To easily present immunotherapy in this review, harnessing T-cell responses against cancer, we divide it into five different classes: immunomodulators (checkpoint inhibitors), cell-based immunotherapies, vaccines, antibody-based targeted therapies and oncolytic viruses [1]. 

Anti-CTLA4 antibody and PD-1 inhibitor therapies as immunomodulators are applied to a wide range of tumor types. While CTLA-4 blockade has been a successful treatment mainly in metastatic melanoma, anti-PD-1/PD-L1 therapies are providing the greatest outcomes in other malignancies, such as non-small cell lung (NSCLC) [152,153], Hodgkin’s lymphoma [154], Merkel-cell carcinoma [155], triple-negative breast cancer [156], renal cell carcinoma [157], urothelial bladder [158,159] and squamous cell carcinoma of the head and neck [160]. Interestingly, patients treated with the PD-1 inhibitor nivolumab start to develop antigen-specific T-cell clones 2–4 weeks after ICI treatment, suggesting that PD-1 signaling blockade promotes anti-tumoral T-cell-dependent activities that enhanced tumor antigen-driven priming of first resident and then circulating T-cells [161]. Those results are particularly relevant since they highlight a window of opportunity to intercept therapeutically relevant TCRs in the peripheral blood of treated patients even if specific issues need to be tackled before reaching clinical practice.

For example, a consensus on the characteristics of peripheral blood TCR repertoires associated with therapeutic responses is still missing since clonotypes differ between blood, which includes the majority of T-cell clones unrelated to the antitumor immune response [65], ascites and tumor [162]. Moreover, CD4+ and CD8+ tumor infiltrating lymphocytes (TILs) should be separated prior to TCR sequencing to avoid a bias due to a higher clonality exhibited by CD8+ TILs compared to CD4+ [163,164], so the clonality in TILs could be due to the high numbers of CD8+ cells [165]. Specific T-clonotypes involved in the recognition of tumor neoantigens via the analysis of TIL repertoires have been found, so the characterization of TCRs in TILs as both biomarkers and therapeutic molecules seems possible [166,167,168]. A recent study by Valpione and colleagues where the prognostic role of TILs’ TCR diversity is examined for several cancers from the Cancer Genome Atlas (TCGA) data heads in this direction [169]. In particular, TCR clonality of TILs pre-treatment can be used as a predictive biomarker for immunotherapy activity and efficacy in metastatic melanoma patients [169].

The presence of neoantigen-reactive T-cells has been assessed across different cancer histologies, such as lung cancer [170,171], bladder cancer [172], head and neck cancer [173,174], ovarian cancer [175,176,177,178], pancreatic cancer [179,180] and gastrointestinal epithelial malignancies [181,182,183,184].

The use of deep sequencing to explore the heterogeneity of tumor cells, especially in solid tumors, allows today to map the hierarchy of clones in the tumor mass and to identify putative oncogene and mutation addicted pathways in tumor cells [149,150], such as those involving transcription factor and oncosuppressor protein p53 [185]. Genomic instability, which is a common feature of cancer cells, leads often to the generation of chromosomal rearrangements and aneuploidy. Many are the examples today among gene fusions and mutations that promote cell transformation in different oncological settings. Those events lead to the unique opportunity to generate neoantigens that could be presented at the cell surface in an HLA-restricted manner [186]. Solid tumors can create an immunosuppressive environment that disfavors adaptive immune responses, thereby preventing recognition. This does not mean that the immune system could not produce immune receptors able to recognize tumor cells but rather that specific immune cells were unfit to elicit an effective immune response considering the tumor microenvironment. It is possible to distinguish at least two categories of tumor antigens: tumor specific antigens (neoantigen and viral antigen) and tumor associated antigens (cancer/testis (CT) antigen, overexpressed antigen and differentiation antigen) [187]. The identification and characterization of tumor antigens can lead to the development of therapeutic strategies, such as vaccines and genetically engineered T-cells, particularly CAR-T, which displayed remarkable clinical efficacy in hematological malignancies, with FDA-approved therapies targeting CD19 (Kymriah, Yescarta, Tecartus, Breyanzi) and BCMA (Abecma) [188,189,190]. 

In parallel, a window of opportunity to find therapeutic TCRs via the analysis of the TCR repertoire in cancer patients is also opened. T-cell receptor-engineered T-cell (TCR-T) therapy has the advantage of targeting potentially any antigen, not only those expressed on the surface of cells, such as with CAR-T-cells. Recent clinical studies on TCR-T showed meaningful results in solid tumor patients [191,192,193], with New York esophageal squamous cell carcinoma-1 (NY-ESO-1) being the most frequently targeted, with a relevant clinical response rate in metastatic melanoma and in metastatic synovial sarcoma patients [194,195,196].

To date, the number of identified antigens and associated TCRs in solid tumors is limited due to issues related to: HLA downmodulation antigen escape, TME immunosuppression and HLA-restriction [187]. In addition, TCR-T-cells require complex procedures of genetic manipulation to avoid mispairing with cognate TCRs, and careful testing for toxicity and efficacy in multiple clinical trials. Despite those hurdles, HTS technologies are now mature for being used to derive therapeutic TCRs and to monitor TCR-T expansion and effect on T-cell compartment in patients.

## 7. Discussion

HTS has become a powerful and increasingly accessible analytical tool for the scientific community to investigate complex behaviors and scenarios from a genetic standpoint.

Recently, deep sequencing applied to profile TCR repertoires has led to an ever-increasing knowledge of immune profiles, whose diversity, along with HLA characteristics, is associated with specific pathological conditions (e.g., infectious diseases, autoimmune diseases and cancer) and outcome of treatments. Major steps forward have been made in those directions by both bulk and single-cell sequencing approaches even if, technically, we can only scratch the real diversity of the repertoire at the surface considering the inherent astronomical richness of T-cells inside the whole body. We listed some considerations that must be taken into account before starting a deep sequencing workflow based on the experimental question one wants to answer and highlighted that the combination of more analytical strategies can lead to more comprehensive information. This additional investigative capacity represents a challenge in data analysis, underlining the need for advanced softwares capable of coupling the information obtained from paired single-cell gene expression and TCR repertoire data [14].

The improvement and diffusion of HTS technology have made it possible to use TCR sequencing worldwide thanks to the development of commercial kits and workflows that allow to sequence samples in a day. The choice of the kit or service depends on the specific goals, and, here, we provided, on one hand, an overview of the current applications of TCR repertoire sequencing and a description of the main providers for this technology on the market.

The vast landscape of services and kits reported in this review provides researchers with a snapshot of what the market currently offers in the area of TCR repertoire sequencing at a cost that is certainly not insignificant. Obviously, TCR repertoire sequencing can be performed in any laboratory equipped with a PCR and sequencer through customized protocols, which are well-detailed and readily available online [197], based on a first step of library preparation combined with a sequencing platform and software data analysis, then representing a more economical choice compared to using kits or services from companies [8,198]. However, it must be considered that handling such a workflow requires an awareness of both experimental biases and bioinformatic data analysis since raw sequencing data require proper data analysis and specific expertise in this field.

Relying on the services of a company is a standardized option as both the workflow and reagent provided are robust and quality controlled, thus likely to be more reproducible rather than in an unspecialized laboratory, plus data could be easily compared with a well-established and validated database that can be provided by the same companies. Using a commercial kit in order to standardize the first part of the process and have support in trouble-shooting could represent a good compromise to solve, for example, library prep biases and then sequence the samples in the lab, considering also that some companies are available to provide even just support in the bioinformatic analysis phase.

Most of the proposed assays are based on the Illumina platform, which currently represents a good compromise between reads length, depth of analysis and bias profile for most experimental designs, while the Thermo Scientific approach relies on Ion Torrent, a technology that poses some problems both for the efficiency of the ligation procedure during library preparation and for the inaccuracy of sequencing of nucleotide homopolymers above 6 units (long repetitions of the same nucleotide) [199]. Further, Ion Torrent and other platforms, such as Roche-454 and Pacific Biosciences, present biases regarding insertions and deletions of nucleotides, which represents an issue in the TCR repertoire sequencing context since an incorrect reading of the CDR3 sequence can lead to erroneous experimental conclusions [1].

To infer protocols regarding reproducibility, replicability and sensitivity, Barennes et al. applied nine commercial and academic TCR sequencing protocols on the same bulk T-cell sample, concluding that method-specific repertoire profiles were largely consistent among replicates and, therefore, each protocol has unique biases when capturing the TCR repertoire [5,47]. 

Several strategies for experimental validation of TCR recognition are available; we reported as an example the TCR Power [2], a tool based on spike-in TCRs that estimates the detection probability of a disease-relevant TCR sequence in an experiment through TCR detection power calculated as a function of TCR frequency, TCR sample count, sequencing depth and read cut-off.

PairSEQ represents a combinatorics-based strategy of computational inference of αβ-TCR pairing [200,201] using multiple sequencings of the same sample and combinatorial analysis [200]. It requires a large number of cells from a given clone to allow chain pairing, thus limiting its application to large samples and highly represented clones [7].

Likewise, other algorithms have been developed, such as ALPHABETR, which allow to determine CDR3α/CDR3β pairs, dual α-TCR clones and clones that share CDR3α or CDR3 β sequences, and to estimate clonal frequencies [201].

In addition to the experimental validation of TCR recognition, bioinformatic tools, such as TCRMatch [202], have been designed to predict binding based on similarity with TCRs, with known epitope specificity and present in the Immune Epitope Database (IEDB). TCRMatch uses TCR β-chain CDR3 sequences, identifies TCRs with a match in the IEDB and reports the specificity of each match. 

The creation of wide collections of TCR repertoires coupled to clinical and biological parameters will allow to derive meaningful therapeutics and complex diagnostics based on multianalytes (TCR signatures) in the future. Future challenges, we believe, will be to develop bioinformatic and AI-based tools to interrogate those datasets and extrapolate clinically relevant and useful information.

In this perspective, the deployment of the TCR repertoire will allow to recognize individuals exposed to parasites or diseases, improve clinical diagnosis and therapeutic choice, identify new targets useful for the development of vaccines and test and monitor vaccine efficacy in inducing a sequence-specific TCR clonal response. The use of gene-engineered T-cells with specific TCRs obtained through this analysis will likely become an impressive immunotherapeutic tool in the near future. The technology is already set and clinical trials will show the feasibility of this therapeutic strategy.

Linking individual T-cell clones with their gene expression profiles leads to an improved immunological interpretation of the TCR repertoire, and combining single-cell TCR sequencing with DNA-barcoded peptide-MHC multimer technology allows a high-throughput identification of antigen-specific T-cells [203,204,205].

A step forward in the analysis of TCR recognition and HLA identification has recently been made by Adaptive Biotechnologies with the development of the immunoSEQ HLA Classifier, building on the work of Emerson et al., which compares TCR sequences to a database of TCRs known to be associated with a certain HLA type, providing information about the possible susceptibility to or protection from disease. However, while this method is restricted to samples analyzed via ImmunoSEQ^®^, more broadly applicable methods are likely in development, as recently published for the YAMTAD system [206].

Besides technical improvements, there is opportunity for combining the technologies discussed here with other tools to understand deeply the association of the TCR repertoire with diseases at the individual and personalized level.

## Figures and Tables

**Figure 1 ijms-23-08590-f001:**
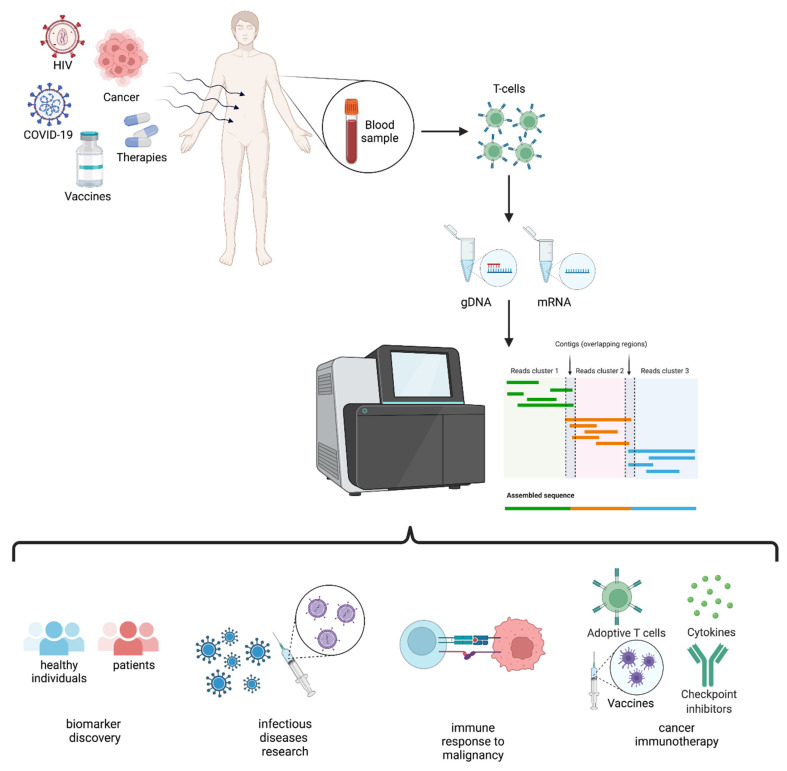
Schematic representation of TCR repertoire analysis and applications.

**Figure 2 ijms-23-08590-f002:**
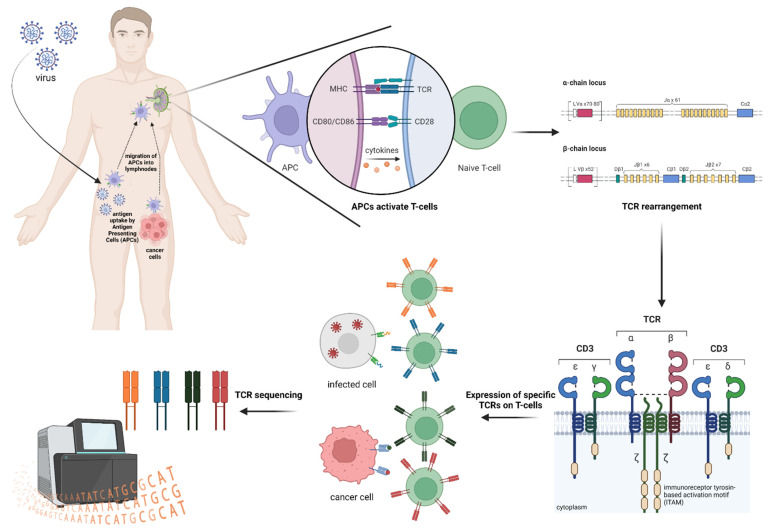
Schematic representation of TCR repertoire generation upon exposure to infectious agents and cancer neoantigens.

**Table 1 ijms-23-08590-t001:** Mechanisms involved in the generation of TCR diversity.

Sources of α/β TCR Diversity	
Recombination of the T-cell α-genes on chromosome 14 and T-cell β-genes on chromosome 7 by RAG1/2 enzymes.	Total α-genes combination: 2392 Total β-genes combination: 1248 Total αβ-genes combination: 2,985,216 [6].
Theoretical diversity by pairing of different in-frame α- and β-chains plus junctional diversity by terminal deoxynucleotidyl transferase activity [6].	10^15^–10^61^ [14]
Experimental diversity evaluation by deep sequencing.	10^4^–10^6^, based on the amount of the sample.

**Table 2 ijms-23-08590-t002:** Examples of association linking HLA type and disease.

Type of HLA Alleles Association	HLA Typing Future Opportunities	Example
With specific infectious diseases or the severity of infection	To provide insight into differences in T-cell repertoires in infectious disease and patterns of T-cell targeting	Heterozygous individuals progress less rapidly to AIDS than HLA homozygous individuals after HIV infection [31]. Kaslow et al. found that HLA-B27 and B57 were strongly associated with slow progression to AIDS [36].
With increased risk of or protection from various autoimmune disorders	To clarify a subject’s disease state and potentially stratify patients for treatment studies.	Association of the HLA class I region has been detected for several autoimmune diseases (AIDs); some examples are: - HLA-B with type 1 diabetes (T1D) [37]; - HLA-C with multiple sclerosis (MS) and Graves’ disease (GD) [37]; - HLA B-27 with ankylosing spondylitis (AS) [38]; - HLA-DRB1, in particular HLA-DRB1*04 and *10 alleles [39] in rheumatoid arthritis (RA); - HLA-G with Crohn’s disease (CD) [40].
With cancer therapy outcomes	To understand and infer the efficacy of immunotherapy in specific individuals	Higher heterozygosity in HLA has been linked to a better response to anti-cancer treatments [41].

**Table 3 ijms-23-08590-t003:** DNA-based vs. RNA-based approaches, choosing the right starting material for TCR profiling.

**Advantages**	
**gDNA**	**mRNA**
easier to obtain; very stable [49]; no requirement for reverse transcription (RT); better reflect the number of analyzed cells; accurate measurement of clonality without bias caused by variable expression levels in different cells.	higher number of copies in a single cell; large information at the gene transcription level; reduced interference of non-coding signals after the splicing process [50]; overall length sequence in the CDR region is easily available; non-productive receptor transcripts are underrepresented [51]. close proximity of V and C regions after the splicing process facilitates PCR amplification [13].
**Disadvantages**	
**gDNA**	**mRNA**
higher concentration input; potential annealing of primers for multiple binding sites; presence of introns and “unused” segments in the sequence of interest that have to be amplified, causing challenges during PCR process [13]; Detection of all the TCR sequences whether they contribute to a productive or a nonproductive segment arrangement [13].	introduction of errors during retrotranscription [52]; easily degraded; high requirements for extraction, transportation and storage.

## Data Availability

All data are reported within the text.

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
