# Peer review of "T-Cell Receptor Repertoire Sequencing and Its Applications: Focus on Infectious Diseases and Cancer"

_ijms, 2022, doi:10.3390/ijms23158590_

Round 1
Reviewer 1 Report
IN the current review by Mazzotti L et al., authors have nicely summarized
overview of TCR repertoire sequencing strategies with respective to infectious diseases and cancer. The review is well written and easy to follow. However, the initial part of the review is engaging from second point (TCR and HIV) it loses the grip. This part needs improvement in language, and text modifications. Although authors have focused on HIV (chronic viral infection) and cancer, there are many other key infectious diseases such as HBV, HCV, and autoimmune diseases which are kept out of focus. Authors are encouraged to briefly describe the scope of these diseases in this context.
While cell therapy is reshaping the current cancer treatment however treating solid tumors successfully with cell therapy is still a huge challenge at this moment. Can authors also describe the use of TCR sequencing in such challenging hard to treat cancers? What is the scope of such methods in gene therapy in addition to neoantigen sequencing analysis?
I will highly encourage to touch base on these components or provide some references which can be highly useful for broader research community in utilizing and exploring the applications of TCR sequencing.
Minor points:
365-368: It is a long and complex sentence. Please simplify it and add reference
The TCR and HIV part needs significant improvement in grammar and sentence structure. Most of the sentences are written in active voice. Please improve.
449-453: This entire part needs to be re-written.
Getting the manuscript edited by professionals for English and sentence formation will highly improve the quality of the review.
Reviewer 2 Report
Lucia Mazzotti and colleagues present a quality and well-written review manuscript focused on T-cell receptor repertoire sequencing and its applications with regards to infectious diseases and cancer.
In this manuscript authors provide an updated overview of TCR repertoire sequencing strategies, providers and applications to infectious diseases and cancer to guide researchers' choice through the multitude of available options. They envisage that the possibility of extending TCR repertoire to HLA characterization will be of pivotal importance in the near future to understand how specific HLA genes shape T-cell responses in different pathological contexts and will add a level of comprehension that was unthinkable just a few years ago.
In this manuscript authors cover such aspects as TCR repertoire analysis, TCR and HLA, TCR repertoire via HTS, as well as TCR and HIV, SARS-CoV-2, cancer.
Authors argue that the deployment of TCR repertoire will allow to recognize individuals exposed to parasites or diseases, improve clinical diagnosis and therapeutic choice, identify new targets useful for the development of vaccines and test and monitor vaccine efficacy in inducing a sequence specific TCR clonal response. The use of engineered T-cells with specific TCRs obtained through this analysis will likely become an impressive immunotherapeutic tool in the near future. The technology is already set and clinical trials will show the feasibility of this therapeutic strategy.
Finally, authors conclude that besides technical improvements, there is opportunity for combining the technologies discussed here with other tools to understand deeply the association of TCR repertoire with diseases at the individual and personalized level.
Overall, the manuscript is highly valuable for the scientific community and should be accepted for publication after the corrections are made.
==============================
Other comments:
1) Please check for typos throughout the manuscript.
2) Lines 584-587. With regards to targeting varioues types of cancers authors are kindly encourages to cite the following article that describes using T cell immunotherapy for treatment of cancers with mutations in tumor supressor genes. DOI: 10.3389/fimmu.2021.707734
